# Novel Antiretroviral Therapeutic Strategies for HIV

**DOI:** 10.3390/molecules26175305

**Published:** 2021-08-31

**Authors:** Rita F. Cunha, Sandra Simões, Manuela Carvalheiro, José M. Azevedo Pereira, Quirina Costa, Andreia Ascenso

**Affiliations:** 1Drug Delivery Research Unit, Research Institute for Medicines, iMed-ULisboa, Faculty of Pharmacy, Universidade de Lisboa, 1649-003 Lisboa, Portugal; r.cunha@campus.ul.pt (R.F.C.); ssimoes@ff.ul.pt (S.S.); mcarvalheiro@ff.ulisboa.pt (M.C.); 2Host-Pathogen Interactions Unit, Research Institute for Medicines, iMed-ULisboa, Faculty of Pharmacy, Universidade de Lisboa, 1649-003 Lisboa, Portugal; miguel.pereira@ff.ulisboa.pt (J.M.A.P.); quirina.c@ff.ulisboa.pt (Q.C.)

**Keywords:** HIV, clinical trials, novel antiretrovirals, vaccines, advanced transdermal nanodelivery systems

## Abstract

When the first cases of HIV infection appeared in the 1980s, AIDS was a deadly disease without any therapeutic alternatives. Currently, there is still no cure for most cases mainly due to the multiple tissues that act as a reservoir for this virus besides the high viral mutagenesis that leads to an antiretroviral drug resistance. Throughout the years, multiple drugs with specific mechanisms of action on distinct targets have been approved. In this review, the most recent phase III clinical studies and other research therapies as advanced antiretroviral nanodelivery systems will be here discussed. Although the combined antiretroviral therapy is effective in reducing viral loading to undetectable levels, it also presents some disadvantages, such as usual side effects, high frequency of administration, and the possibility of drug resistance. Therefore, several new drugs, delivery systems, and vaccines have been tested in pre-clinical and clinical trials. Regarding drug delivery, an attempt to change the route of administration of some conventional antiretrovirals has proven to be successful and surpassed some issues related to patient compliance. Nanotechnology has brought a new approach to overcoming certain obstacles of formulation design including drug solubility and biodistribution. Overall, the encapsulation of antiretroviral drugs into nanosystems has shown improved drug release and pharmacokinetic profile.

## 1. Introduction

### 1.1. Global Epidemiology and Viral Genome of HIV

The human immunodeficiency virus (HIV) is still a very prominent disease worldwide. Acquired Immunodeficiency Syndrome (AIDS) can now be considered a chronic infection since patients are living longer due to the several options of antiretroviral therapy [1]. The number of new cases has decreased, but nevertheless, according to the latest UNAIDS global statistics, 38.0 million people are living with HIV in 2019 and 1.7 million became newly infected with HIV in 2019 [1,2].

The HIV genome is formed by two identical single chain RNA molecules, which are confined to the core of the virus particle. As all retroviruses, HIV capsid includes a reverse transcriptase (RT), responsible for the transcription of the viral genomic RNA (gRNA) into the proviral (double stranded) DNA (dsDNA), which can be integrated into the host cell genome due to the function of another viral-encoded enzyme, the integrase (IN).

The viral genome is delimited at both ends by long-term repeat repeats (LTR). The viral gene transcription is followed by the group-specific antigen (*gag*) gene, which codes for the outer core matrix protein (MA/p17), the capsid protein (CA/p24), the nucleocapsid (NC), and the nucleic-acid stabilizing protein. The *gag* gene is followed by the polymerase (*pol*) gene, which is responsible for the production of RT, the RNase H, and IN. The envelope (*env*) gene codes for two membrane glycoproteins gp 41 (transmembrane protein, TM) and gp120 (surface protein, SU) which is composed of five variable regions (V1 to V5) and five conserved regions (C1 to C5) [3].

### 1.2. Replication Cycle of HIV

The initial step of HIV replication cycle involves SU binding to the host protein CD4, present in T helper lymphocytes, macrophages, and dendritic cells. The interaction between the CD4 binding site (CD4 bs) present in SU subunit with CD4 causes conformational changes in the SU glycoprotein that exposes the coreceptor binding site [3,4]. Binding to coreceptor induces further conformational changes in the TM subunit, disclosing the fusion peptide which will connect to the target cell membrane due to its extremely hydrophobic nature, thereby enabling the fusion of both viral envelope and host cell membrane [4]. Fusion of viral envelope could occur directly with plasma membrane or alternatively with endosome membrane after endocytosis. Following fusion, the viral capsid is then released in the cytosol forming the reverse transcriptase/pre-integration complex (RTC/PIC) that includes the two copies of gRNA, the viral proteins CA, NC, IN, RT, and Vpr as well as cellular proteins, namely cyclophilin A [5]. Inside this structure, RT converts the single-stranded gRNA into a double stranded DNA (dsDNA). The RTC/PIC is transported through the cytoplasm into the host cell nucleus [5,6] via the nuclear pore, where RTC/PIC interacts with proteins of the nuclear pore complex, namely, Nucleoporin 358 and Nucleoporin153 [5,7]. The structural integrity of RTC/PIC seems to be maintained until it enters the nucleus, minutes before the dsDNA integrates into host cell chromosome [5,6]. The integration process is mediated by the integrase protein (IN) starting by removing nucleotides from the 3′ ends of the proviral DNA, and then, proceeding to catalyze a nucleophilic attack to the phosphodiester bonds of the DNA chains, thus forming a covalent bond between viral and host DNA. This is an essential step in viral replication allowing the establishment of latently infected cells [8,9,10]. Some cell transcription factors enhancers bind to LTR and the regulatory proteins such as Tat, Rev, and Nef are produced. HIV-1 transcription from the LTR promoter is activated by the Tat protein through interaction with the nascent *trans*-acting-responsive RNA hairpin structure [9,10]. The Env glycoproteins, after translation, processing, and cleavage by cellular furin protease migrate to the plasma membrane. Meanwhile the MA domain targets the Gag-Pol polypeptide to budding sites at the host cell membrane where it interacts with cytoplasmic tail of TM subunit of Env. This event allows virion assembly with gRNA and all the other structural and accessory viral proteins and budding of immature viral particles [11]. During or soon after budding, the viral protease (PR) cleaves the Gag-Pol polypeptide which will allow the release of all structural proteins, such as MA, CA, and NC, as well as their reorganization into mature virions [9,10].

### 1.3. Viral Phases of HIV

The period between the infection of the first host cell and the detection of the virus in the blood is called the eclipse phase and usually lasts from 7 to 21 days. After the infection of the first cell, the virus continues to replicate in the mucosa, submucosa, and adjacent lymphatic tissue. The replication concentrates in the gut-associated lymphatic tissue (GALT) quite early [12,13]. Then, it follows an exponential rise of the viral loading in which the CD4+ cell counting rapidly decreases. This phase is characterized by flu-like and non-specific clinical signs that usually last between 7 to 10 days. After a few weeks, the immune system can generate a response [9,14]. The cellular immune response starts with the activation of CD8+ cytotoxic lymphocytes. Their T-cell receptor (TCR) will bind to viral proteins, which are in turn connected to the antigen-presenting molecule (MHC I) to eliminate the infected cells [9]. Generally, after 3 to 5 weeks, a humoral response starts to produce specific neutralizing antibodies that will destroy the virions via phagocytosis. The convergence of both types of immune responses leads to a decrease in viremia and a new rise in CD4+ cell count. The period, in which there is an infection but without antibodies, is called the “serological window period” [14]. Even though this is an asymptomatic phase, and the viral loading is somewhat controlled, there is still a loss of immune cells since the virus continues to replicate in the lymphatic tissue (its reservoir), destroying its structure. Then, the viral loading becomes higher as the CD4+ cell count diminishes, leading to the beginning of the AIDS stage. In this stage, the patients are more susceptible to opportunistic infections [14].

Several aspects make it difficult to eradicate the virus once a patient is infected. One of them is related to the absence of proofreading activity in the viral RT, causing a great number of mutations and genetic diversity in the HIV genome. The other one is concerned with the ability of the virus to infect resting memory or naïve cells, leading to a latent viral state [9,14]. The problem with viral latency is that it can occur even after patients have undergone antiretroviral therapy reducing viremia to an undetectable level [9].

Several drugs with specific mechanisms of action on distinct targets in the replication cycle of HIV have been approved so far. In this review, these antiretroviral drugs including those from the most recent phase III clinical studies will be here discussed besides presenting other promising research therapies such as advanced antiretroviral nanodelivery systems.

## 2. Antiretrovirals and Therapeutic Targets

Based on the replication cycle of HIV, several therapeutic targets and antiretroviral drugs have been developed over the years. Nowadays, initial therapy regimens have at least three different drugs. In general, antiretrovirals can be classified into seven major types, according to their mechanism of action [15,16]:-Nucleotide/Nucleoside Reverse Transcriptase Inhibitors (NRTI);-Non-Nucleotide Reverse Transcriptase Inhibitors (NNRTI);-Integrase Inhibitors (II);-Protease Inhibitors (PI);-Fusion Inhibitors (FI);-Pharmacokinetic Enhancers (PE);-CCR5 Antagonist.

Numerous examples of these antiretroviral drugs are listed in Table 1.

### 2.1. Nucleotide/Nucleoside Reverse Transcriptase Inhibitors (NRTI)

The first drugs to be developed were the Nucleotide/Nucleoside Reverse Transcriptase Inhibitors (NRTI). Their structure is very similar to the viral nucleosides, except absence of a hydroxyl group in the 3′ position of their deoxyribose sugar, thereby preventing a phosphodiester bond between the NRTIs and the next 5′ nucleosides. As a result, the nucleoside chain is interrupted and the proviral DNA is not formed. These drugs are formulated and delivered as prodrugs, which need to be phosphorylated to become active [15,18]. NRTIs are classified as competitive inhibitors of the RT, as they bind to the enzyme active site being integrated into the viral DNA chain [20].

Regarding pharmacokinetics, this class of drugs is known to have a long intracellular half-life period, good oral bioavailability, and no restrictions of administration, and improbable interactions with other drugs [17]. On the other hand, few side effects have been documented such as myelosuppression, pancreatitis, and neuropathy. However, the resistance mechanisms have been noticed as well [15,17]. The first NRTI approved was Zidovudine (AZT) in 1987, and the last one was Emtricitabine in 2003 [30].

### 2.2. Non-Nucleoside Reverse Transcriptase Inhibitors (NNRTIs)

The Non-nucleoside Reverse Transcriptase Inhibitors (NNRTIs) bind directly to the RT in a non-competitive inhibition process [16]. NNRTIs bind to a specific pocket of the viral RT, away from the active site, thereby inducing a conformational change that inhibits the enzymatic activity [20]. NNRTIs do not inhibit the RT of any other retroviruses, nor one of HIV-2 [18]. NNRTIs are quite cheap to produce and allow the single tablet regimens [19]. These drugs can cause rashes, nausea, vomiting, fatigue, mood swings depression, jaundice, conjunctivitis, and respiratory issues [16]. Just like the NRTIs, NNRTIs are very prone to inducing resistance [18]. The first NNRTI to be approved was Nevirapine (NVP) in 1996, the same year when combined antiretroviral therapy (c-ART) was first introduced [17]. The most recent drug to be approved was Doravirine in November 2018 [31].

### 2.3. Integrase Inhibitors (IIs)

In the last decade, the viral integrase has been successfully used as a therapeutic target for HIV through a unique biochemical mechanism. The integrase inhibitors designed until now are strand transfer reaction inhibitors (INSTIs). These inhibitors are quite selective drugs since they interact with the co-factors (metallic cations) at the active site of this enzyme and then with the complex formed by viral DNA and integrase. The pharmacophore binds to the metallic cations whereas a lipophilic group interacts with the viral DNA-integrase complex [18].

INSTIs can cause hypersensitivity reactions, rash, jaundice, dark-colored urine, nausea, vomiting, fatigue, blisters in the mouth and skin, diarrhea, and loss of appetite [13]. Raltegravir (RAL) was the first integrase inhibitor approved in 2007.

The most recent approval was Elvitegravir in 2014 [19]. However, its marketing authorization in the European Union was withdrawn in 2016 for commercial reasons [32].

### 2.4. Protease Inhibitors (PIs)

Protease inhibitors bind to this enzyme in a competitive manner as its natural substrate. The inhibition of this enzymatic activity leads to immature virions and less viral spreading [18,20].

The main inconvenience of these antiretroviral drugs is that the protease gene is very prone to mutation, which can easily lead to resistance [18].

PIs are known to cause arrhythmia, heartburn, fatigue, jaundice, dizziness, abdominal pain, mouth sores, kidney stones, and dark-colored urine [16]. Saquinavir (SQV) was the first PI to be approved in 1995 by the Food and Drug Administration (FDA) and in 1996 by the European Medicines Agency (EMA) [31,33]. The most recent approval for this drug category was Darunavir in 2006 by the FDA and in 2007 by the EMA [30,31].

### 2.5. Fusion Inhibitors

Enfuvirtide (Fuzeon^®^) was the first drug approved as a fusion inhibitor (gp41). The gp41 component can be sub-divided into the N-terminal heptad repeat (NRH)-binding domain which is analogous to Fuzeon^®^, and a pocket-binding domain (PBD), with hydrophobic pockets that bind to the NHR exterior [26]. This drug acts by inhibiting the connection between NHR (or HR-1) and the C-terminal heptad repeat (CHR or HR-2), thereby preventing the formation of the 6 helix-bundle (6HB) or “core”, and ultimately, the membrane fusion with the viral envelope [26].

Enfuvirtide was approved in 2003 as a treatment option for patients infected with HIV-1 who were in an advanced stage of disease progression and were resistant to other antiretroviral drug classes [28]. It is administered by subcutaneous injection and the standard regimen is 90 mg twice a day [29]. The most common side effects are reactions near the injection site, nausea, diarrhea, fatigue, and eosinophilia [29].

The TORO clinical trials demonstrated few benefits in the administration of enfuvirtide in cases where patients have shown resistance to therapy with NRTIs [29]. The viremia and CD4 cell counts were compared in patients taking both enfuvirtide and an optimized conventional ART regimen versus only the conventional ART regimen. After 48 weeks of treatment, 30% of the enfuvirtide patients had their viremia reduced to <400 copies/mL versus 12% in the conventional therapy arm. At a 95-week follow-up, 17.5% of patients taking enfuvirtide had their viral loading under 50 copies/mL [29].

Despite being a promising therapeutic option, it presents certain disadvantages. Firstly, it has a very short half-life, which leads to an inconvenient posology and administration route, and consequent low patient compliance. Secondly, it has a very low antiretroviral activity and only active against HIV-1. Finally, this drug has a very low threshold for resistance [27,34]. The resistance occurs mainly due to mutations in the HR-1 region of gp41, even though mutations in HR-2 are also possible [17,29].

To surpass these issues, several enfuvirtide-based lipopeptides have been synthesized over the years including LP40, LP46, LP52, and LP80 as second-generation fusion inhibitors. Although LP46 presented higher antiviral activity than enfuvirtide, its phase II clinical trial was interrupted due to reports of serious side effects [35]. LP52 was designed based on LP46 and LP40 exhibiting a higher threshold for drug resistance and capable of inhibiting resistant strains. Additionally, it has a longer half-life and higher antiviral activity with low IC50 values (pM range) [35]. In a recent study [36], several new lipopeptides were developed using fatty acids with different lengths, for example, LP80 which was found to be a quite potent fusion inhibitor against resistant strains. It showed similar cytotoxicity compared to LP52 and enfuvirtide in different cell lines, such as TZM-bl cells, HEK293T cells, and MT-4 cells. Other promising results were obtained in pharmacokinetic studies in healthy rhesus macaques [36]. When administered at 3 mg/kg to rhesus macaques chronically infected with simian-human immunodeficiency virus (SHIV), once a day for two weeks, viral loading was reduced to below the detection limit after 4 days of treatment in 3 out of 5 monkeys [36].

### 2.6. Pharmacokinetic Enhancers

Cobicistat was approved in 2012 as a pharmacokinetic booster in co-administration with other antiretrovirals. Contrarily to ritonavir, it has no specific antiretroviral activity. However, it is substantially more selective than ritonavir, considering that it inhibits only the CYP3A isoenzyme subfamily, leading to fewer drug interactions [24,25]. By inhibiting the CYP3A isoenzymes, cobicistat allows an increase in the plasma concentrations of other antiretrovirals, such as PIs and NNRTIs, thereby enabling higher intervals in administration with a lesser pill burden and improved adherence to therapy [25].

Some clinical trials showed an increase of serum creatinine levels and a decrease in glomerular filtration rate (GFR) in patients taking cobicistat. Nevertheless, in a phase I study, these abnormalities were explained due to a decline in the activity of certain transporters, such as SLC22A2, which are essential for the elimination of creatinine. This means that cobicistat causes higher serum creatinine levels, thereby altering the GFR estimation calculated using the Cockcroft-Gault formula. Thus, it does not really alter GF [24,25]. It is now known that this drug restrains some cation renal transporters. However, no dosage adjustments are required for patients with defective liver or kidney function [24]. Regarding CYP3A4 inducers, such as carbamazepine and rifabutin, cobicistat is not as effective when administered simultaneously with these drugs, since it is mostly metabolized by CYP3A4 and CYP2D6 [24].

### 2.7. CCR5-Antagonist

CCR5-antagonists were the first class of antiretroviral drugs to target host cells and not virions. In fact, CCR5 is the most common co-receptor present in most target cells in the early stage, which allows it to be a very promising therapeutic target for that stage [21].

The CCR5-antagonist Maraviroc (MVC) was approved by both FDA and EMA in 2007 [31]. MVC binds to the hydrophobic pocket of the co-receptor inducing a conformational change that avoids its recognition by viral gp120, thereby limiting virion entry into peripheral blood mononuclear cells (PBMC) [15,37]. MVC is mostly metabolized in the liver by CYP450 enzymes, such as CYP3A4 and CYP3A5, which is why it should be administrated cautiously with other CYP450 inducing or inhibiting drugs, such as PIs [37]. The most common adverse effects are nausea, diarrhea, fatigue, and headaches [22]. Furthermore, it is known some mechanisms of resistance related to a higher affinity of gp120 towards CCR5 [37].

The combination antiretroviral therapy (c-ART) could be considered as a winning strategy to assume a therapeutic regimen consisting of drugs with different targets. Fixed-dose combination of ARVs administered as a single tablet approved until date is listed in Table 2. This combination was first implemented in 1996, and since then, some disadvantages were found as the interference of PIs and NNRTIs with the metabolization process of other common drugs [25]. Ritonavir started to be progressively co-administered with other first-generation PIs, such as lopinavir or saquinavir, and afterward, with second-generation PIs, including atazanavir and darunavir [25]. As ritonavir could constrain the metabolization process, higher concentrations of other antiretrovirals and larger intervals in their administration were possible to achieve.

## 3. New Drugs—Overview of Phase III Clinical Trials and Recent Approvals

Nowadays, the wide range of therapeutic options and c-ART regimens have allowed that AIDS become a chronic disease with lower mortality [44]. Although c-ART is the best therapeutic strategy until now, it is far from being the ideal solution considering that patients are forced to life-long medication, which presents a considerable risk of resistance emergence and rebound in viral loading. Moreover, these patients also suffer from numerous side effects [45,46].

When discussing a cure for the infection of the HIV virus, it is important to establish the difference between a “sterilizing cure” and a “functional cure”. A “sterilizing cure” consists of the elimination of both actively infected cells and the virus latent reservoir, whereas a “functional cure” implies a prolonged suppression of viremia to an undetectable level and sustaining a normal CD4+ cell count [44].

There are currently several new drugs and vaccines being tested in pre-clinical and clinical trials for HIV treatment or prophylaxis. Hereby, the most recent clinical studies on phase III will be further discussed.

### 3.1. Post-Attachment Inhibitors: Ibalizumab

Ibalizumab (Tograzo^®^) granted a “breakthrough therapy status” in 2015, and it was approved in March 2018 by the FDA [47]. In September 2019, the EMA approved Ibalizumab for use in patients with multi drug resistant HIV when conventional therapeutic combinations failed to be effective [48]).

As an IG4 monoclonal antibody (IG4 mAb), ibalizumab connects to domain 2 of the CD4 receptor and leads to conformational changes which avert the gp 120—co-receptor interactions, thereby preventing the viral entry [49]. This innovative drug presents a very low ADCC (antibody dependent cellular cytotoxicity) and hence a quite low rapport towards cytotoxic immune responses. Therefore, CD4+ cells will not be destroyed through cytotoxic pathways [49].

Ibalizumab is the first drug to target the CD4 receptor and therefore the first one to be considered a post-attachment inhibitor [47,50]. The FDA recommends the subcutaneous administration of 2000 mg loading dose, and subsequently, 800 mg maintenance dose every 2 weeks [47]. The half-life of this drug is estimated to range between 3 and 3.5 days [23,49]. Ibalizumab is generally well tolerated, with some adverse effects such as rash, headaches, dizziness, nausea, and diarrhea [48,49].

In summary, ibalizumab presents as a very promising and safe therapeutic option for patients suffering from resistance to the recommended ARV regimens, and it might be more convenient considering the administration intervals, thus allowing a higher adherence to therapy and patient autonomy [23].

### 3.2. Long-Acting Injectable Cabotegravir/Rilpivirine Formulation

Cabotegravir (CR) (Vocabria^®^) is an integrase strand transfer inhibitor (NSTI) structurally like dolutegravir [51,52]. It was approved by the EMA in December 2020 [53]). This drug is manufactured as a once-daily oral tablet and as an intramuscular (IM) and subcutaneous (SC) injectable. Cabotegravir is formulated as a long-acting (LA) suspension for monthly to quarterly injection dosing. Prior to the initiation of the injections, cabotegravir (Vocabria^®^) and rilpivirine (Edurant^®^) oral tablets are taken for approximately one month to assess tolerability to the medicines. Then, cabotegravir IM injection (Vocabria) is used in combination with rilpivirine IM injection (Rekambys^®^) as a complete long-acting regimen dosed once monthly or once every 2 months [52,54]. The half-life of this drug is particularly long, ranging between 20 to 40 days [52]. CR is strongly bound to albumin and eliminated through the liver, even though no dose adjustments are required for patients with impaired hepatic function [52]. In turn, rilpivirine is an NNRTI with a long half-life too ranging from 30 to 90 days. It might also interfere with liver and pancreatic enzymes and cause headaches, nausea, dizziness, and fatigue [52]. Rilpivirine was also aproved by the EMA in December 2020 [55].

Recently, the results from Phase III clinical trials (FLAIR and ATLAS) of this novel formulation have been published [54]. FLAIR was an open-label Phase III trial in which the included patients were ART-naïve. Aside from K103N, none of them presented any other NNRTI resistance-associated mutations (RAMs) [52]. This trial was divided into two phases. The induction phase lasted for 20 weeks in which all patients took the oral fixed-dose combination of abacavir/lamivudine/dolutegravir. At 20th week, the maintenance phase began, and all patients with less than 50 copies/mL were randomly distributed to a group that remained with the initial therapy and to another one that started a different oral therapy regimen with 30 mg cabotegravir + 25 mg rilpivirine once a day for 4 weeks. Following the 4 weeks, an intramuscular injection of 600 mg cabotegravir + 900 mg rilpivirine was administrated, and afterward, a maintenance dose of LA cabotegravir 400 mg + 600 mg LA rilpivirine once a month [52,56,57]. After 48 weeks, the results showed that 7 patients out of 283 taking oral therapy (2.5%) had over 50 copies of viral RNA/mL as opposed to 6 patients out of 283 taking the LA formulation (2.1%). However, there were patients in both arms of the trial that withdrew from the study due to ineffectiveness of the treatment and few others confirmed the virologic failure in keeping viral loading under 200 copies/mL. Nevertheless, these results were enough to prove the non-inferiority of LA injectable therapy compared to oral therapy [50,52,56,58]. Patients were generally satisfied with this new therapy regimen and injection site reactions (ISRs) were mostly mild and constrained [52].

ATLAS was also an open-labeled and randomized trial, in which all the participants had less than 50 copies/mL of HIV RNA and followed an oral therapy regimen with 2 NRTI + 1 INSTI, NNRTI or PI for a minimum of 6 months. The patients were randomly distributed to continue with their usual regimen or take the LA therapy. Similarly, to the FLAIR study, the patients received 30 mg LA cabotegravir + 25 mg LA rilpivirine for 4 weeks. Afterward, an intramuscular injection of 600 mg of LA cabotegravir + 900 mg LA rilpivirine was administered as a loading dose, and then, a maintenance dose of 400 mg LA cabotegravir + 600 mg rilpivirine was taken once a month through the same route of administration [52,59,60]. The results showed that after 48 weeks only 5 out of 308 patients doing LA therapy had over 50 copies of viral RNA/mL (1.6%), compared to 1.0% (3/308) in the control group. This means that LA cabotegravir + rilpivirine therapy was considered non-inferior to the triple oral therapy. Contrary to the FLAIR study, there was a higher incidence of adverse reactions [50,52].

The LA injectable therapy has demonstrated significant advantages when compared to oral therapy in both studies. Although most patients found this option preferable, there are still some concerns about this treatment option, such as the viral resistance, the compliance to this medication schedule, and how to proceed when an injection is missing [52]. Additionally, it is known that rifampicin activates the metabolism of CR, reducing its concentration in the plasma. This might cause a problem in HIV-positive patients co-infected with Tuberculosis since rifampicin is commonly used to treat it [52].

Despite these promising results, there are thus still some questions to be answered, including the effect of LA therapy on pregnant women. Now, there are other ongoing studies, for example, MOCHA that aims to provide information regarding the pharmacokinetics and safety of using LA CR on adolescents from 12 to 18 years old and ALTAS 2M in which a different regimen of 600 mg LA CR + 900 mg LA rilpivirine is taken every 8 weeks [52].

In 2021, the FDA approved cabotegravir and rilpivirine to be used as an association of drugs named Cabenuva^®^. This was the first HIV-1 treatment regimen where the administration occured only once a month. Cabenuva^®^ is composed by two separate long-acting injectable suspensions of cabotegravir and rilpivirine. This regimen also includes the administration of Vocabria^®^ 30 mg tablets and rilpivirine 25 mg tablets (Edurant^®^) prior to the injections to ensure that the long-acting formulations are well tolerated [61,62,63].

### 3.3. Fostemsavir

Fostemsavir (Rukobia ^®^) was developed as a prodrug of temsavir, an attachment inhibitor. Its mechanism of action is innovative, as it connects to the viral glycoprotein gp120 which in turn is unable to bind to the CD4 receptor and, therefore, compromise the viral entry [51,64,65]. The BRIGHTE study was a Phase III clinical trial that assessed the efficacy of this new drug. The included participants were infected with multi-resistant HIV-1 and with a viral loading over 400 copies/mL [65,66]. Patients were assigned into two cohorts: one randomized (RC) and another non-randomized (NRC). In the RC, patients took the placebo or 600 mg fostemsavir twice a day along with their failing ART drugs for 8 days. From the 9th day forward, all patients received 600 mg fostemsavir twice daily along with optimized background therapy just like the patients in the NRC [65,66]. After 24 weeks, 146 out of 272 patients in the RC (54%) had less than 40 copies of viral RNA/mL. In the NRC, the percentage of patients who had their viral loading less than 40 copies/mL was 36% (36/99) [65,66]. The results also showed a low prevalence of adverse effects, being headaches, nausea, and diarrhea the most frequent ones [65,66]. However, few patients experienced more severe adverse effects, such as acute renal failure and hepatocellular injury, among others [65]. Fostemsavir did no present major interactions with other classes of ARV drugs [65].

Rukobia^®^ was approved by the EMA in February 2021 to be used in patients with multi-drug resistant HIV-1. One of its most severe side effects is the immune reconstitution inflammatory syndrome [67]. Additionally, some interactions with other drugs were identified., such as rifampicin, anti-epilepsy drugs, like carbamazepine and phenytoin as well as antineoplasic drugs [67].

### 3.4. Leronlimab (PRO 140)

Leronlimab (PRO 140) is a humanized IgG4 monoclonal antibody. Its target is the CCR5 chemokine receptor since it binds to the extracellular domains of this molecule, i.e., the loop 2 and the N-terminus [68,69]. By this way, PRO 140 prevents virions from entering and infecting other host cells [68,69]. There are many similarities between Maraviroc and PRO 140 [68], and their difference in binding sites may suggest that these drugs could act by a synergistically way [70].

There are ongoing phase IIb/III trials aiming to evaluate the safety and effectiveness of leronlimab. In CD 02 trial, the efficacy of leronlimab has been tested in patients with an unsuccessful ART regimen. They took 350 mg of PRO 140 subcutaneously or placebo for 1 week along with their regular medication. One week later, participants took PRO 140 with an optimized regimen. Unfortunately, the results were not promising since the failure rate was 76% due to the emergence of X4 viral strains for which PRO 140 was ineffective [68]. CD 03 is another clinical trial, in which participants are still being recruited. This trial intends to test the transition of stable patients treated with conventional therapy into a weekly regimen of subcutaneous leronlimab in monotherapy, by assessing the viral loading for 48 weeks [71].

Overall, high tolerability and high threshold to the resistance of this new drug might allow it to be suitable for pre- and post- exposure prophylaxis [68].

### 3.5. UB-421

UB-421 is an Fc-glycosylated humanized IG1 antibody that targets the CD4 receptor. It prevents the attachment and viral entry by attaching to domain 1 of that receptor and blocking it in a competitive manner [72].

The safety and efficacy of UB-421 have been assessed in several clinical trials. One of them is a phase II/III trial that started in September 2019 whereupon UB-421 was assessed along with a failing ART regimen at an initial stage for 1 week, and later, UB-421 was evaluated with an optimized background regimen for 24 weeks. The primary outcome measures were the viral loading log10 difference from the baseline [73]. Additionally, a phase III study has just started in January 2020 with the main goals of evaluating the efficacy, safety, and tolerability of this drug as a monotherapy agent. Participants were distributed into two cohorts. Cohort 1 consisted of a control group taking standard c-ART, while in cohort 2 patients took only intravenous UB-421 at 25 mg/kg bi-weekly for 26 weeks. Afterward, all patients entered a follow-up phase receiving the standard c-ART. The primary outcome measure was the number of patients without virologic failure [74].

### 3.6. Others

Islatravir (NRTTI) and lenacapavir (capsid inhibitor—new drug class) are potential first-in-class drugs in late-stage clinical trials, with significant clinical results to date. Both drugs have long half-lives, stability, and high genetic barrier to resistance, thus demonstrating activity at low doses in clinical studies. This clearly supports their development as a combination regimen in long-lasting formulations, both oral formulations and injectable forms.

Besides the new drugs, several attempts to develop an effective and safe vaccine against HIV have been also made as mentioned in future perspectives [75,76,77,78,79,80,81,82,83,84,85].

## 4. Novel Therapeutic Strategies

### 4.1. New Transdermal Drug Delivery Systems

Since most ARVs are administered orally, it is important to identify the main adversities of this route of administration, including those related to bioavailability, frequency of administration, and the hepatic first-pass metabolism. Transdermal drug delivery systems (TDDS) are currently being investigated to diminish those issues [86,87,88,89]. The TDDS present an alternative to the conventional oral ARV regimens since this administration system allows obtaining a controlled and continuous drug release by avoiding the pharmacokinetic variations of the oral administration. This means that it may be possible to use drugs with a short half-life and find a simpler and acceptable regimen for patients [86,87,89].

Besides the reservoir and the drug itself (i.e., drug concentration, pka, molecular weight (ideally around 500 Da), log P (preferably ranged between 1 and 3), and the melting point) [86,87,88], a TDDS should have other essential components, as follows: (a) permeation enhancers to increase the permeability through the stratum corneum (e.g., pyrrolidones, surfactants, phospholipids, and solvents); (b) pressure-sensitive adhesives (PSA) to keep contact between the skin and the device; (c) a release liner to cover the patch, and (d) a backing layer which should not allow the diffusion of any excipients [86,87].

#### 4.1.1. Transdermal Delivery System for Tenofovir Alafenamide

Tenofovir Alafenamide (TAF) is an NRTI commonly used for HIV treatment as part of oral therapy regimens. A controlled matrix delivery system was recently developed for this drug due to its low oral bioavailability [88]. In this system, the active substance is dispersed in a polymer matrix and a suitable solvent, which later evaporates, forming a drug reservoir. Then, the reservoir is shaped and interconnected with several layers. The overall system is composed of an adhesive layer that controls the release rate of the drug and it is in contact with the skin, followed by the drug reservoir and a second adhesive layer, connected to an exterior impermeable laminate [86,87,88]. Accordingly, this study [88] aimed to formulate a patch able to continuously release 8 mg/day TAF for one week. For that proposal, silicone-based or polyisobutylene (PIB) suspension patches and acrylate solution patches were used, and several formulations were prepared varying the components, TAF concentration, and permeation enhancers for each patch. Specific parameters were determined, such as (a) crystallization of TAF in each matrix, (b) effect of TAF particle homogenization, (c) coat weight and TAF amount in each patch after exposure to stress conditions, and (d) in vitro skin permeation studies using Franz diffusion cells. Overall, the suspension silicone patch formulated with 15% TAF (*w*/*w*), silicone, oleic acid, oleyl alcohol, and mineral oil as permeation enhancers revealed to be the most suitable since it presented the target flux permeation rate of 7 µg/cm^2^/h through a 50 cm^2^ patch area for an entire week, reaching a daily dose of 8.4 mg TAF. Additionally, this formulation was stable over time, non-irritating to the skin, and convenient when peeled off [88]. Despite these positive results and the promising chance of a new route of administration for TAF, further studies are still needed regarding both pharmacokinetic and safety of this new device [88].

#### 4.1.2. Transdermal Delivery of Enfuvirtide (T20) via Ultrasounds

As previously discussed, enfuvirtide (T20) is an entry inhibitor, usually administered by subcutaneous injection, 90 mg twice a day, which enables it rather inconvenient in terms of patient compliance. Thus, its transdermal delivery could be quite interesting option. However, there are some obstacles, including the high molecular weight (4.492 Da) of this drug, which could compromise its diffusion through the stratum corneum [87,90]. Therefore, the ultrasound technique could be a suitable way to surpass this problem, since it reduces the barrier function of the stratum corneum [91,92]. Ultrasound waves are known to create pores that allow large molecules to cross the epidermis besides contributing to a fluid state of lipid skin layers, which in turn, promotes the transcellular pathway [91,93].

A recent study has assessed the effects of transdermal delivery of T20 using a low-frequency and low power ultrasound transducer patch in porcine models. The models were divided into 3 separate groups for 30 days as follows: one control group receiving injectable T20 twice a day, another group undergoing ultrasound treatment with saline solution, and a third one treated with transdermal T20 via ultrasounds [90]. In this last case, T20 was in direct contact with the skin. The final device was obtained using wound dressing patches and a silicone ring that served as a reservoir, over which the ultrasound transducer was placed [90]. All groups were evaluated to understand the differences between them regarding skin health and bioavailability. The skin health criteria were based on histologic cuts and trans-epidermal water loss. The bioavailability of T20 was evaluated through liquid chromatography/electrospray ionization mass spectrometry (LC-MS/MS) [90]. Overall, no significant differences between the saline group and the transdermal T20 group were observed, which indicates that ultrasound did not affect the skin. The histologic cuts showed mild signs of inflammation in the active patch group. In addition, the animals from the active patch group had a longer Tmax and a lower Cmax when compared to the injectable T20 group. The plasma concentrations were generally lower with the transdermal treatment as expected [90].

### 4.2. Nanosystems for Drug Delivery

Nanotechnology has contributed to several applications in drug delivery through different routes of administration and overcoming certain formulation obstacles, such as solubility, bioavailability, and drug stability [94]. Additionally, nanosystems could be a promising strategy for targeted therapy since they allow the encapsulation of drugs or specific genes that could be transported not only to infected cells but also to reservoir tissues, including the central nervous system and lymph nodes, thereby potentially eradicating the virus [94].

There are several types of nanosystems that could function as carriers for ARV drugs, such as liposomes, niosomes, solid-lipid or polymeric (e.g., PGLA) or diamond nanoparticles, and dendrimers, among others (Figure 1) [94]. In general, liposomes and niosomes are less toxic and cost-effective. Both are made of an aqueous compartment surrounded by a lipid bilayer and could be a promising alternative for drug delivery as these carriers can be easily absorbed by macrophages [95]. For example, stavudine was already encapsulated into gelatin nanoparticles, which in turn were incorporated into liposomes. Accordingly, several formulations were successfully prepared in this report and characterized in terms of drug release, cytotoxicity, and hemocompatibility [95].

Dendrimers are somewhat toxic but have numerous reactive groups capable of forming conjugates for targeted drug delivery [94,96]. Dendrimers possess a typical three-dimensional branched structure where the outer layers are appropriate for conjugation, while the inner layers are quite effective for drug encapsulation, resulting in a controlled drug release [96]. In a recent study published in 2013, zidovudine was encapsulated into poly (propyl ether imine) dendrimers to surpass the short half-life and provide a more continuous release of this drug [97]. The study showed that not only this delivery system reduced the hemolytic effect, but it also prolonged the drug release decreasing the occurrence of side effects [97]. In another study, carbosilane dendrimers (G3-S16 and G2-NF16) were used to encapsulate zidovudine, efavirenz, and tenofovir [98]. The encapsulated drugs were tested for antiviral activity in PBMC cells and TMZ-bl cells infected with X4 and R5 viral strains. The results showed an enlarged antiviral activity of all three drugs when formulated with dendrimers [98].

Polymeric nanoparticles, such as quitosan and poly (lactide co-glycolide), are known to be very effective in drug delivery while exhibiting low toxicity like metal gold or silver nanoparticles [94]. Nanoparticles have been tested to improve ARV delivery to the central nervous system since many ARV drugs suffer efflux mechanisms, which may contribute to the spreading of this virus, and thereby, to the development of HIV-related neurologic disorders [99,100]. For example, transferrin is one of the nanoparticles studied for this purpose due to the abundance of transferrin receptors in the blood–brain barrier [94,99]. In a recent study, nanodiamond particles were studied to load efavirenz (EFV) and deliver it to the brain [101]. This study compared nanodiamond particles (ND) with both unmodified and modified surface (ND-COOH and ND-NH2) in terms of toxicity and drug loading capacity. ND-COOH was found to be less suitable than the other two since it induced a higher production of reactive oxygen species (ROS) [101]. On the contrary, the formulation with unmodified nanodiamond particles (ND-EFV) presented a suitable and slower release profile through a blood–brain barrier model, and it was able to impair viral replication for a longer period in comparison with free EFV [101]. Overall, this study suggests that ND particles are a promising drug delivery system, due to their nontoxic nature and ability to cross the blood–brain barrier. However, further studies still need to be performed to evaluate the effect in in vivo models and possible side effects, since ND particles may interfere with the expression of genes related to neuronal function [101]. Another recent study attempted to demonstrate the effects of using PGLA nanoparticles loaded with EFV and saquinavir (SQV) [102]. In general, ARV formulated with PGLA nanoparticles showed lower IC50 values in comparison with the free ARV drugs. The release profile of these nanoparticles was also quite favorable, since the drugs were first rapidly released, and then, at a continuous rate. Additionally, when adding free tenofovir to these nanoparticles, a synergic effect was obtained resulting in dose reduction to impair HIV activity [99]. In another report, PGLA nanoparticles were loaded with raltegravir (RAL) and EFV, and further incorporated in a thermo-sensitive vaginal gel for HIV prophylaxis [103]. The goal was to prepare a gel that would acquire a gel-like texture at 37 °C and liquid at room temperature. The PGLA nanoparticles were here prepared via a modified emulsion-solvent evaporation method and characterized through multiple experiments, including in vitro release studies with human cervical cells. When compared with an EFV-RAL solution, the loaded nanoparticles had a lower EC90 and were able to ensure a constant release of these drugs, despite their different intracellular concentrations and routes of metabolization [103]. Overall, this formulation was considered a successful and promising option for pre-exposure HIV prophylaxis [103]. Still regarding prophylaxis, cellulose acetate-phthalate (CAP) nanoparticles have shown promising outcomes when incorporated into thermosensitive gels. Although most used as a coating agent for other formulations, CAP was found to possess ARV activity by promoting viral disintegration and interfering with the mechanisms of viral entry. Moreover, CAP was stable at low pH, which facilitated a vaginal drug delivery [104,105]. In another study [105], a thermosensitive gel was formulated using EFV-loaded CAP nanoparticles. This formulation was assessed for cytotoxicity and prophylactic activity in human cervical cells (HeLa) and TZM-bl cells against an EFV solution. The results revealed a remarkable encapsulation efficacy as well as lower cytotoxicity in HeLa cells treated with EFV-CAP nanoparticles formulated in the thermosensitive gel. Moreover, these nanoparticles showed enhanced prophylactic activity (at 5 ng/mL) in TMZ-bl cells when compared with the EFV solution [105].

Similarly, dolutegravir (DTG) loaded CAP nanoparticles were incorporated into a thermosensitive gel and tested at pH = 4.2 and pH = 7.4 to simulate vaginal and seminal fluid conditions, respectively [104]. In this study, the pH clearly influenced both the drug release and the cytotoxicity of the tested formulation [104].

A novel tenofovir alafenamide fumarate (TAF) nanofluidic implant developed by Pons-Faudoa et al. [105,106] revealed to be potential as a subcutaneous delivery platform for long-term pre-exposure prophylaxis to HIV transmission according to in vivo data (SHIV-Challenged Nonhuman Primates). It had similar effects as oral TAF dosing with a lower dose [105].

In summary, the examples of the most recent in vitro studies of ARV loaded nanocarriers for HIV management are listed in Table 3. Overall, the use of long-acting ARVs with alternative drug delivery systems contributes for the long-term sustained release of these drugs with a controlled kinetic profile which certainly accounts for the success of these new ARVs therapies. In addition, the drug delivery systems might also provide more convenient dosing, enhance tissue penetrance, improve viral resistance profile, and reduce drug toxicity [105].

## 5. Conclusions and Future Perspectives

HIV infection is still considered one of the major pandemics worldwide. In the 1990s, with the emergence of the first antiretroviral drugs and c-ART, the status of this infection changed from deadly illness to chronic infection, allowing infected people to live as long as non-infected individuals. However, c-ART presents many downsides, such as side effects, frequency of drug administration, and the possibility of viral resistance, while it still does not provide a definitive cure.

Several studies have been made to shift the therapeutic strategies towards this infection. Accordingly, new drugs from the already well-known ARV drug classes and others with a novel therapeutic target have been evaluated in clinical trials. Moreover, some classical ARV drugs have been also studied with a different route of administration and/or formulation, enabling a prolonged drug release as well as surpassing some compliance issues derived from polymedication in HIV patients. In fact, nanotechnology has been shown to provide a more targeted and controlled release of antiretrovirals overcoming several formulation obstacles including cytotoxicity.

The eminent need for an effective vaccine against HIV has been also discussed over recent years. Despite various attempts, this goal has not been achieved yet. To this date, there have been few effective clinical trials for vaccines against HIV. The target for neutralizing antibodies is Env; thus, this could be a suitable immunogen to be considered when designing a vaccine. A phase III clinical trial will be set to assess the efficacy of a new vaccine containing mosaic antigens. This might be an innovative challenge to formulate a “global vaccine” and overcome the major obstacle of the genetic diversity of HIV.

Nevertheless, further investigation is urgently needed especially regarding prophylactic vaccines and pharmacokinetic/pharmacodynamic in silico prediction followed by well-defined in vivo studies of new antiretroviral drugs.

## Figures and Tables

**Figure 1 molecules-26-05305-f001:**
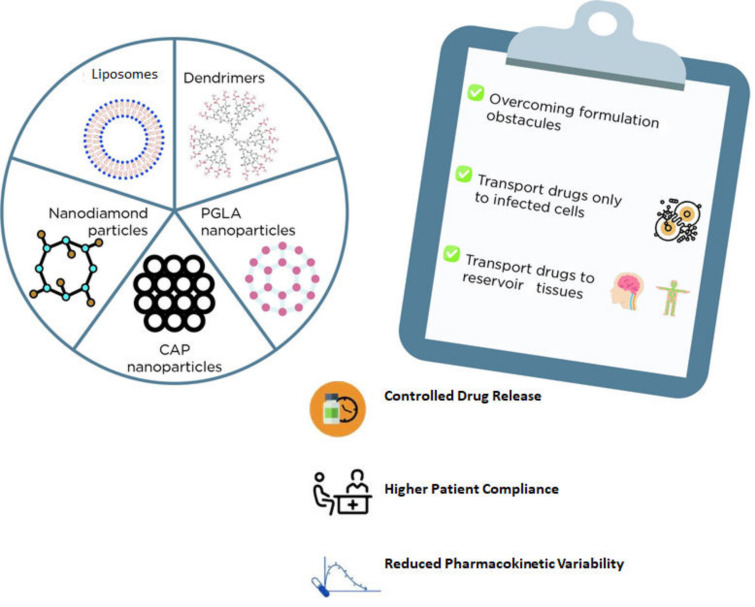
Examples nanosystems for drug delivery and the main advantages of their usage.

**Table 1 molecules-26-05305-t001:** Classification of antiretroviral drugs for HIV and therapeutic advantages and disadvantages.

Class of Antiretroviral Drugs	Therapeutic Target	Approved Drugs	Advantages	Disadvantages	Ref.
Nucleotide/Nucleoside Reverse Transcriptase Inhibitors	Reverse Transcriptase	Abacavir (ABC) Ziagen^®^	Long intracellular half-life period High oral bioavailability Few interactions No problems with administration	Highly prone to resistance Adverse effects: myelosuppression neuropathy pancreatitis, nausea, vomiting fatigue, anemia, lactic acid accumulation	[16,17,18]
		Tenofovir disoproxil fumarate (TNF) Viread^®^			
		Lamivudine (3TC) Epivir^®^			
		Emtricitabine (FTC) Emtriva^®^			
		Zidovudine (AZT) Retrovir^®^			
Non- Nucleotide Reverse Transcriptase Inhibitors	Reverse Transcriptase	Efavirenz (EFV) Sustiva^®^	More selective than NRTIs Cheaper to produce Single-tablet regimens	Very prone to resistance. Rash, nausea, vomiting, fatigue, mood swings, depression, jaundice, conjunctivitis, and respiratory issues	[17,19,20]
		Nevirapine (NVP) Viramune^®^			
		Delavirdine (DLV) Rescriptor^®^			
		Etravirine (ETR) Intelence^®^			
		Rilpivirine (RPV) Edurant^®^ Doravirine (Pifeltro^®^)			
Integrase Inhibitors	Viral Integrase DNA integration)	Raltegravir (RAL) Isentress^®^	Very selective drugs, they interact with two components of viral replication	Hypersensitivity reactions, rash, jaundice, dark-colored urine, nausea, vomiting, fatigue, blisters in the mouth and skin, diarrhea, and loss of appetite	[16,17,19]
		Dolutegravir (DTG) Tivicay^®^ Bictegravir Biktarvy^®^			
Protease Inhibitors	Viral protease protein synthesis	Ritonavir (RTV) Norvir^®^	Active against HIV-1 and HIV-2	High prevalence of resistance. Arrhythmia, heartburn, fatigue, jaundice, dizziness, abdominal pain, mouth sores, and urinary tract issues	[17,19,21]
		Nelfinavir (NFV) Viracept^®^			
		Atazanavir (ATZ) Reyataz^®^			
		Darunavir (TMC114) Prezista ^®^			
		Saquinavir (SQV) Invirase^®^ Fortovase^®^			
CCR5 antagonist	CCR5 co-receptor	Maraviroc Celsentri^®^	Effective in cases of resistance to conventional therapy regimens	Only effective in R5 viruses Nausea, Diarrhea, Fatigue, headache	[22,23]
Post-attachment inhibitors	CD4+ cells	Ibalizumab Trogarzo^®^	Pharmacokinetics allows for a weekly administration Does not cause CD4 depletion No evidence of resistance	Immune reconstitution inflammatory syndrome	[17,24]
Pharmacokinetic Enhancers	CYP3A subfamily	Cobicistat Tybost^®^	More selective than ritonavir Less drug-drug interactions	Might cause raises in serum creatinine Side effects in the gastrointestinal tract	[17,25,26]
Fusion Inhibitors	gp41 subunit	Enfuvirtide Fuzeon^®^	Decreases viral load in a c-ART regimen Increases CD4 cell counts Low toxicity High specificity	Short half-life Low threshold for drug resistance Can cause reaction in its injection-site, nausea, and fatigue High cost Inconvenient route of administration	[18,27,28,29,30]

**Table 2 molecules-26-05305-t002:** Fixed-dose combination of ARVs administered as a single tablet approved until date according to literature and EMA information.

Approved Drugs	Active Substances	References
Genvoya^®^ Biktarvy^®^	150 mg elvitegravir/150 mg cobicistat/200 mg emtricitabine/10 mg tenofovir 50 mg bictegravir/200 mg emtricitabine/25 mg tenofovir alafenamide	[38,39]
Atripla^®^	600 mg efavirenz/200 mg emtricitabine/245 mg tenofovir-DF	[40]
Rezolsta^®^	800 mg darunavir/150 mg cobicistat	[40]
Triumeq^®^	50 mg dolutegravir/600 mg abacavir/300 mg lamivudine	[41]
Evotaz^®^	300 mg atazanavir/150 mg cobicistat	[42]
Descovy^®^	200 mg emtricitabine/10 mg tenofovir alafenamide 200 mg emtricitabine/25 mg tenofovir alafenamide	[43]

**Table 3 molecules-26-05305-t003:** Examples of main outcomes from in vitro studies of ARV loaded nanocarriers for HIV management.

Nanocarriers + ARV	Main Outcomes	References
Liposomes + Stavudine	Liposomes were revealed to be a promising alternative for stavudine delivery as these carriers can be easily absorbed by macrophages.	[95]
Dendrimer + Zidovudine	The formulation reduced the AZT hemolytic effect and prolonged the drug release, decreasing the occurrence of side effects.	[97]
Carbosilane Dendrimers+Zidovudine Carbosilane Dendrimers + Efavirenz Carbosilane Dendrimers + Tenofovir	An enlarged antiviral activity of all three drugs was observed when formulated with dendrimers.	[98]
Nanodiamond Particles + Efavirenz	A suitable and slower release profile through a blood–brain barrier model was obtained impairing viral replication for a longer period.	[101]
PGLA nanoparticles + Efavirenz PGLA nanoparticles + Saquinavir	An enlarged antiviral activity of all three drugs was obtained with PGLA nanoparticles.	[102]
PGLA nanoparticles + Efavirenz + Raltegravir [thermosensitive gel]	A lower EC90 and a constant release of these loaded drugs were obtained being a promising option for pre-exposure HIV prophylaxis.	[103]
CAP nanoparticles + Efavirenz (thermosensitive gel)	High encapsulation efficacy and lower cytotoxicity in HeLa cells were observed besides enhanced prophylactic activity in TMZ-bl cells treated with EFV-CAP nanoparticles.	[105]
CAP nanoparticles + Dolutegravir (thermosensitive gel)	pH (4.2 and 7.4) influenced both the drug release and the cytotoxicity of this formulation.	[104]

## Data Availability

Not applicable.

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
