# Peer review of "Novel Antiretroviral Therapeutic Strategies for HIV"

_molecules, 2021, doi:10.3390/molecules26175305_

Round 1

Reviewer 1 Report

 the manuscript has been significantly improved and now warrants publication

Author Response

please find the document in attachment

Reviewer 2 Report

Cunha et al. review new strategies for delivery of antiretroviral therapy.  This is an important topic to cover as there are new approaches in development for administering antiretroviral drugs for treatment of HIV infection and pre-exposure prophylaxis.  The review article is fairly comprehensive and discusses new advances in ART and ART delivery.  However, there is room for both minor organizational and content improvement.

Specific points

  • The abstract nicely summarizes what is covered in the review. However, the Introduction lacks further explanation of the purpose of the review.  While there is considerable detail described about the HIV life cycle, it is not inherently clear why this placed in the Introduction and not its own section where it should set up the following section on ARVs and therapeutic targets.
  • Doravirine is mentioned in section 2.2 but should also be included in the list of ARVs in Table 1. There is no mention of the INI Bictegravir.
  • Other drugs under development of significance that could be discussed: the NRTTI, Islatravir, which has long half-life and stability and high genetic barrier to resistance, and the capsid inhibitor, Lenacapavir.
  • Among preclinical ARV drug delivery systems being tested that could be included in the review is the implantable nanofluidic device (Pons-Faudoa FP et al. 2021 Adv Therapeutics PMID: 33997267, Pons-Faudoa FP et al 2020 Pharmaceutics PMID:33080776
  • It would be worth discussing the use of long-acting ARVs with alternative drug delivery systems for long-term release of ARVs.
  • As the focus is on antiretroviral therapy and delivery, the inclusion of a limited summary of vaccine progress does not appear justified or fitting.

Author Response

This manuscript is a resubmission of an earlier submission. The following is a list of the peer review reports and author responses from that submission.

Round 1

Reviewer 1 Report

The authors propose a review which, after an introductory initial part, reports  the main characteristics of innovative anti-HIV therapeutic strategies, with a particular attention to possible contribution of novel drug delivery systems and nanotechnology to antiretroviral therapy. This is a subject of potential importance in the present and future context of the HIV-1/AIDS  epidemic, which is worth to be considered trying to examine the involved aspects in as much detail as possible.

The proposed review is sufficiently complete and clearly written.  However, in the opinion of this reviewer, some major or minor changes must be made to update the contents of this review, correct certain imperfections, and  further ameliorate its completeness and readability.

Here is the list of the changes to make.

General point.

The present version seems not adequately updated regarding to approved ARVs. In particular,  authors include in paragraph 3, phase III clinical trials, some drugs that have been now approved by FDA or EMA or both, such as ibalizumab, cabotegravir/rilpivirine, fostemavir. Moreover, probably also the term “commercial” is not properly used in case they will distinguish antiretroviral drugs approved and utilized by many years from those more recently approved. Thus, authors must completely revise the present subdivision of the paragraphs 2 and 3, their contents, and their titles in order to avoid confusion of the readers about already approved or not drugs.

Specific points.

Line 31.

Update data and reference 2 (line 610) using the last available UNAIDS 2020 fact sheet.

Line 35.

“The virus particle produces an ezyme..” . Substitute “produces” with “contains” or “includes”.

Lines 113-114.

any initial therapy regimen must have”, better “initial therapy regimens have”. Then, the second part of the sentence “at least three different drugs with distinct therapeutic targets” is not correct: in fact NRTI and NNRTI have the same therapeutic target. Rephrase properly.

Line 115, table 1. 

Authors say that ARVs can be classified into eight major types but the successive list includes only seven groups. Indeed, table 1 includes eight classes of ARVs. In table 1, for NNRTI the column therapeutic target includes also viral replication, but this is a therapeutic target for all ARVs and should be omitted! In any case, according to the “General point” this paragraph and table 1 must be revised.

Line 257, Table 2.

No mention for table 2 is present in the text. Moreover, the title of table 2 could be confusing. In fact,  the term “combination antiretroviral therapy” and its acronym c-ART, firstly mentioned in line 259 without any explanation, should be briefly, but properly, explained somewhere in the text (introduction? beginning of revised paragraph 2 ) as the winner strategy to assume a therapeutic regimen consisting of drugs with different targets, while table 2 refers to the relatively recently approved and used strategy to utilize a fixed-dose combination of ARVs administered as a single tablet.

Line 275.

See “General point”. Approved by EMA in September 2019.

Line 297.

See “General point”. Approved by FDA at the  beginning of 2021, EMA pending.

Line 299-300.

It is not clearly stated that the oral and injectable administrations differ in terms of interval of administration.

Line 352.

See “General point”. Approved by FDA in July 2020, and by EMA at the beginning of 2021.

Reviewer 2 Report

The authors Cunha et al. review the current but also novel antiretroviral therapeutics to treat HIV infection. I do have a couple of comments and suggestions in order to improve the current manuscript:

Major

p. 2 line 54-55: the current knowledge of HIV replication is that HIV capsid is not absorbed by endosomes

p.2 lines 56-57: "Inside the phagocyte, H+ ions are released, causing a decrease in pH value, and inducing the delivery of the capsid into the cytoplasm" - is wrong and not according to the current knowledge of HIV replication.

p.2 lines 57-61: "The viral RT will convert the viral single-stranded RNA into complementary DNA (cDNA), forming a pre-integration complex, which can be integrated into the host cell genome, while the RNA strand is destroyed by RNase H [3]. The cDNA is transported through the cytoplasm and into the host cell nucleus via the nucleopores, with the intervention of the Vpr protein".

This is also not correct anymore and the current view of what happens between entry and integation has drastically changed latest by the publication of Zila et al. (doi: 10.1016/j.cell.2021.01.025.)

p.2 line 70-71 "Tat will activate transcription by connecting its trans-activation response element (TAR) element to the LTR region". This is wrong, as TAR is an element present on the LTR

p.2 line 71-76 "The Gag-Pol protein resulting from the translation of mRNA is transported and becomes embedded into the plasma membrane of the host cell [8]. Meanwhile, all the other structural viral proteins are produced and ensembled together in the plasma membrane, forming a multiprotein structure. This leads to the activation of the viral protease (PR), which will allow the release of all structural proteins, such as MA, CA and NC, as well as their reorganization into mature virions [6,7]." This is wrong. HIV assembles at the plasma membrane, only the envelope glycoproteins become embedded into the plasma membrane. HIV is released as an immature virus, maturation does only take place in the extracellular space.

p.8 line 277: Ibalizumab is already on the market since September 2020 and sold as Trogarzo. IRIS (Immune Reconstitution Inflammatory Syndrome) has been reported as a side effect.

p.9 line 298: Both drugs were already approved by the EU and EMA in December 2020 and are known as Vocabria and Rekambys

p.10 line 352: Fostemsavir (Rukobia) has been approved by Feb 2021

Minor:

P.1 l.44 - Please correct to gp41